# What instruments are available to aid or evaluate personalised care delivery, from the perspectives of healthcare practitioners and service users? A narrative scoping review

**Louise Johnson**[1,2]*, **Beth Clark**[2], **Lyndsay Court**[1,2], **Hayden Kirk**[3], **Matthew Wood**[1,4], **Sharon Jackson**[5], **Mari Carmen Portillo**[2,6]

1 University Hospitals Dorset NHS Foundation Trust, Bournemouth, Dorset, United Kingdom, 2 School of Health Sciences, Faculty of Environmental and Life Sciences, University of Southampton, Southampton, United Kingdom, 3 Hampshire and Isle of Wight Healthcare NHS Foundation Trust, Southampton, United Kingdom, 4 Dorset County Hospital NHS Trust, Dorchester, Dorset, United Kingdom, 5 Portsmouth Hospitals University NHS Trust, Cosham, Portsmouth, United Kingdom, 6 NIHR ARC Wessex, Southampton Science Park, Innovation Centre, Southampton, United Kingdom

* louise.johnson@uhd.nhs.uk

## Abstract

### Background

Although the widespread implementation of personalised care is commonly cited as one of the solutions to managing the burden of increasing multimorbidity, there is no established method for assessing and evaluating personalised care delivery. This review sought to describe the range of existing tools, instruments or methods for assessing, evaluating or measuring personalised care delivery, from the perspectives of healthcare practitioners and/or service users.

### Methods

A scoping review of literature published since 1990.

### Results

From 3851 potential citations, 172 were included. Of these, 103 reported the development of a new instrument, and 69 reported the adaption of an existing instrument. Most instruments (81%) were designed for use in a specific clinical population. A focus on supported self-management was particularly common (80%). Few instruments were identified that explored the views of healthcare staff (n = 8) or carers (n = 1). Content analysis generated six domains: understanding the person; understanding capability; understanding behaviour; personalised care interventions; experience of care; and wider determinants. These domains have been used to propose a concept framework.

**Data availability statement:** All relevant data are within the manuscript and its Supporting Information files.

**Funding:** The author(s) received no specific funding for this work.

**Competing interests:** The authors have declared that no competing interests exist.

**Abbreviations:** SDM, Shared Decision Making.

## Conclusion

This review identified a high number of instruments, designed to support personalised care delivery or evaluation. Many were designed to understand a single construct of care (e.g., supported self-management), at an individual level (e.g., patients) and in a specific population (e.g., diabetes). For clinicians wishing to utilise a standardised instrument for a specific purpose, there are many to choose from. Yet no tools encompass the full spectrum of constructs encapsulated within personalised care. Future work should focus on *how* instruments are used to improve personalised care delivery, particularly through a less siloed, multimorbidity lens.

## Introduction

Developing healthcare systems with the readiness and ability to deliver holistic and personalised care, is a key priority for the UK National Health Service (NHS) [1]. While personalised care interventions hold promise for addressing the challenges of preventing and managing multimorbidity, their success depends on achieving widespread implementation; which remains limited [2]. One factor inhibiting implementation is the lack of an established method for assessing and evaluating personalised care delivery. Robust evaluations not only determine whether an intervention works, but also why and how – enabling us to learn from effective interventions and develop new ones [3]. In personalised care, an effective evaluation should consider the views of multiple stakeholders and would ideally be applicable across diverse settings and populations. This is pertinent given the increasing prevalence of multimorbidity, and the subsequent burden on healthcare utilisation, healthcare expense, overall functioning, and quality of life [4].

Before methods for full scale evaluation of personalised care are developed, it is important to understand the range and characteristics of the existing instruments and methods available to evaluate and/or measure personalised care, or aspects of it. While there are various instruments available in the field of personalised care, there is also a lack of guidance on how to effectively select and utilise these tools. Enhancing this understanding is essential to improving clinical delivery and systematic implementation. This scoping review will compile and categorise the existing instruments, laying the groundwork for a deeper understanding of their role in personalised care delivery, and identifying areas that require further research.

As there is no universally accepted definition of personalised care, our review aligns to the description of whole person care; an approach that considers multiple dimensions of the patient and their context, including biological, psychological, social and possibly spiritual and ecological factors, and addresses these in an integrated fashion that keeps sight of the whole [5]. Implementation of a more personalised approach can be enabled through six evidence-based and inter-linked interventions, as outlined in the NHS Personalised Care Operating Model [6]. These are shared decision making, personalised care and support planning, enabling choice, social prescribing and community-based support, supported self-management and personal/integrated health budgets [6].

Scoping reviews are a form of evidence synthesis, that are used to describe the available literature on a topic, with the specific objective of describing the volume and nature of the existing evidence [7]. Through this review, we aim to identify and describe the available instruments through which personalised care can be evaluated, or through which elements of it may be measured. Given the complexity of personalised care, and the potential breadth and variety of research in this field, a scoping review provides the appropriate methodology to do this.

### Review question

What tools, instruments or methods are available to aid or evaluate personalised care delivery, from the perspectives of healthcare practitioners and/or service users?

### Objectives were to

a) identify the different *types* of measurement and/or evaluation instruments used to understand personalised care delivery within clinical services.

b) identify and describe the *key characteristics* of these methods or instruments.

c) report *how* the methods and instruments are used, and *who* they are used with.

d) summarise existing knowledge relating to the evaluation of personalised care delivery, from the perspective of a range of stakeholders.

## Methods

We conducted a scoping review with a systematic methodology, broadly following the stages outlined by Arksey and O'Malley: identifying the research question; identifying relevant studies; selecting studies; charting data; collating, summarising and reporting data [8]. Our protocol was pre-registered on the OpenScience Framework (osf.io/2wfg7) and we used The Preferred Reporting Items for Systematic Reviews and Meta-Analyses extension for Scoping Review (PRISMA-SR) checklist to guide reporting.

To focus our research question, we used the Participant, Concept, Context (PCC) criteria [9,10].

P: adults with long term condition(s) who are accessing healthcare services; healthcare staff who are involved in the delivery of services to people with long term conditions; healthcare teams, services, or systems of care, that are involved in supporting people with long term conditions ("service units")

C: studies that report the development of a method or instrument aimed at understanding, evaluating or measuring the delivery of one or more element of personalised care.

C: any health care setting, including both physical and mental health

To keep the review broad, we included studies that focus on the evaluation of one or more element of personalised care [11]. Studies were included from all healthcare settings, and all long-term condition patient groups.

### Identifying relevant studies

Studies were identified through searching electronic databases (CINAHL, MEDLINE, AMED and EMBASE), using the terms outlined in S1 File. Due to the vast number of records identified when we used all three PCC categories, we applied a search using the concept search terms only – these related to "personalised care" and "measurement". We applied the participant and context criteria when completing our title and abstract review. The search was run from 1990−11th December 2024 and was restricted to English-language publications.

## Selecting studies

All identified records were imported to Endnote20 for removal of duplicates. Remaining records were transferred to Rayyan [12], to facilitate collaborative title and abstract screening. Titles were screened by the primary researcher (LJ), removing records that clearly did not meet the inclusion criteria. The remaining records were independently reviewed by two members of the research team, through abstract screening and full paper retrieval if necessary. Five reviewers were involved in this process (LC, BC, MW, SJ, LJ), with each paper being considered by at least two. Data labels were agreed a-priori, with new labels generated as required. Discrepancies were resolved by consensus of the whole group. Reasons for exclusion are reported in the PRISMA-SR flow chart (Fig 1)

Studies were eligible if they met the following criteria: a) inclusion of people over the age of 18 with one or more LTCs and/or healthcare staff or services that support people over 18 with one or more LTCs; b) reports on the development of a

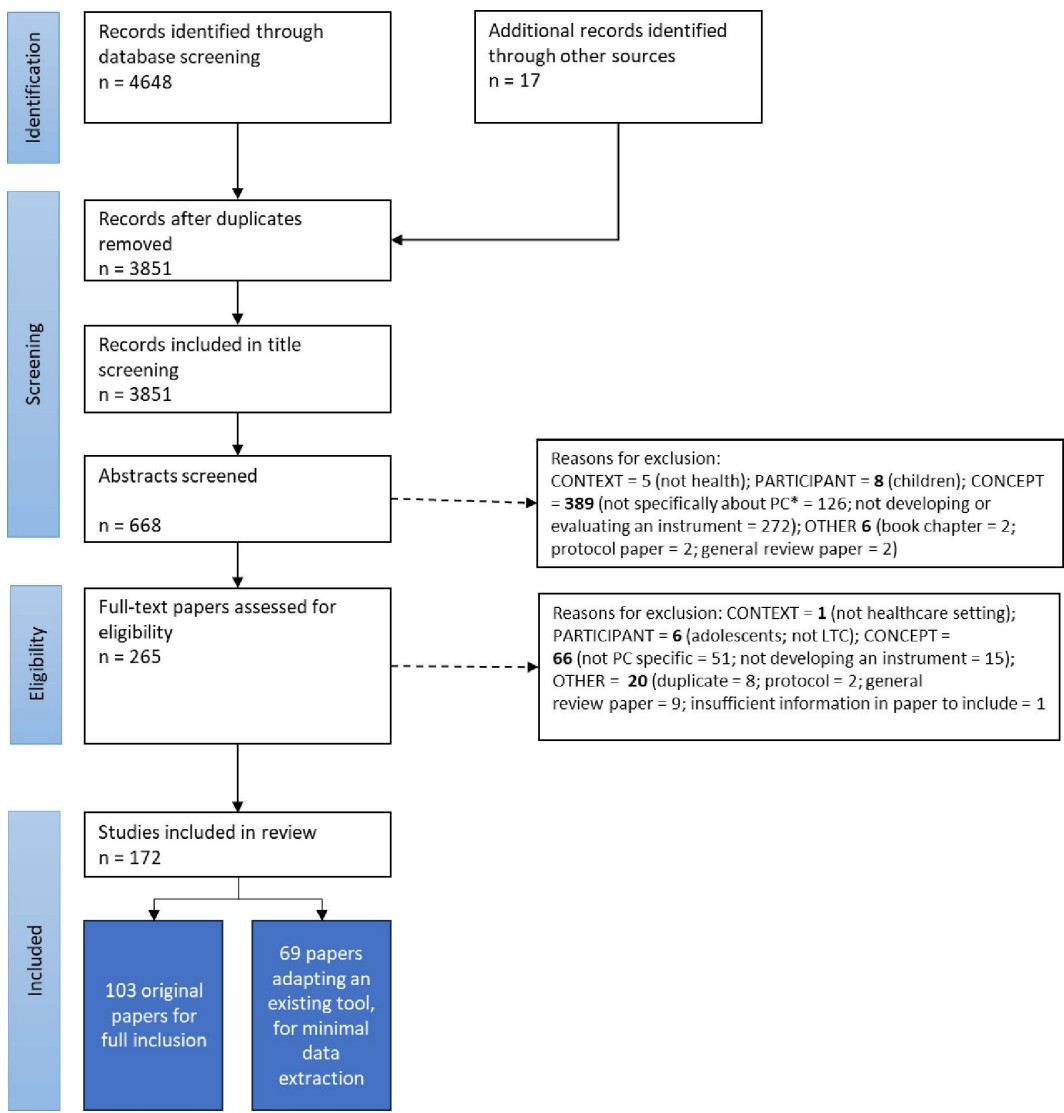

**Fig 1. PRISMA Flow Chart.**

method or instrument for understanding, evaluating or measuring personalised care delivery; c) the evaluation/measurement instrument is (or could be) used within clinical practice (e.g., a survey); d) written in English.

Papers were excluded if they: developed and tested an *intervention* for personalised care (including Decision Support Tools); focussed solely on paediatric to adult transition services; used an instrument within a study, but did not develop or evaluate the instrument itself; used qualitative methods to collect data relating to personalised care, but not with the intention of those methods being used outside of the research. Instruments that sought to specifically and solely understand personal characteristics, such as self-efficacy, coping behaviours or activation, were excluded from the review. Whilst we recognise these are important characteristics that influence personalised care, they also play an independent role in overall health outcomes, beyond personalised approaches. To maintain a direct focus on personalised care delivery, we only included instruments that explored two or more characteristics, with the explicit intention of informing personalised care delivery. Given the significant number of related constructs and the high volume of instruments in these related fields, this was also important to maintain a realistic scope for the review. Further studies were identified by reviewing the reference lists of selected papers.

### Charting data

Data extraction followed the principles outlined by Pollock et al [13]. A data extraction template was developed, trialled with the first five papers, and refined through an iterative process involving multiple members of the research team (LJ, MCP, BC). Once finalised, data relating to study characteristics and findings was extracted and charted by a single researcher (LJ). Throughout the process, members of the research team met regularly to discuss and reflect on the findings, and to collectively address areas of ambiguity or concern. New data items were incorporated throughout the review process. Where necessary, authors of studies were contacted to clarify aspects of their research or provide missing data.

In addition to extracting data from each paper, we reviewed the instruments themselves. Using conventional qualitative content analysis [14,15], we describe the nature of the instruments, and other factors relating to concept [13]. Each item within each instrument was coded, using categories that were derived directly from the instruments themselves. This initial process of coding was led by the primary researcher (LJ). Codes, illustrated by examples, were then discussed with the wider research team to agree categorisation and mapping, to aid simplification [10].

It was not our intention to scope the effectiveness of these tools nor their psychometric properties – and therefore this information was not extracted. Furthermore, in line with accepted recommendations for scoping reviews, we did not apply any formal critical appraisal tool to the included studies [10,16].

### Collating, summarising and reporting results

Key findings were tabulated and are presented in a descriptive way, as per the initial study objectives. Data relating to the study characteristics and the study findings is reported as a narrative summary, providing a broad overview of the evidence in this field. Similarities and differences across papers are described, alongside an overview of the strengths and limitations of the evidence base. Mapping of codes from the content analysis is presented narratively, and visually as a concept framework.

### Results

Our original search returned 3851 titles for review. After title screening, 668 abstracts were screened for inclusion, 265 full text papers were reviewed, and 172 were included in the final review. Of these, 103 papers underwent full review. The remaining 69 papers reported the adaption of an existing personalised care instrument; these underwent partial review, to describe the nature of these adaptions. In total, 101 unique personalised care instruments were identified (S2 File).

## Overview of study characteristics

The number of publications in this field has risen steadily since 1990, demonstrating an ongoing interest and drive for instruments that can support personalised care delivery (Fig 2).

Whilst most research took place in North America, Europe and parts of Asia, we identified studies from a wide range of geographical settings. The development of novel instruments took place almost exclusively in high income countries, likely reflecting a greater focus on personalised care within these settings, particularly within health system policy. The adaption of existing instruments involved a broader range of settings, with a large proportion being adapted for use in Asia (42%), followed by Europe (35%) and North America (14%). Most of these papers report language (22%) and cultural adaptions (45%), with associated psychometric testing. Other adaptations included testing within a new clinical population (16%), a new clinical setting (4%), or the modification of the instrument itself (e.g., addition or removal of items) (13%).

**Participants.**  The majority (94%) of instruments were designed to understand the perspective of patients or service users; primarily using self-administered questionnaires. We identified only a small number of instruments that were developed to evaluate the perspectives of healthcare practitioners (n = 8) or caregivers (n = 1). Two instruments included mirrored questions, to directly compare patient and provider perspectives of shared decision-making consultations [17,18]. Healthcare practitioner focussed instruments used observational methods to evaluate practitioner-patient interactions [19], or self-report to understand perceived behaviours [20], knowledge, confidence and attitudes of clinicians themselves [21].

Instruments have been developed and tested with a wide demographic of patients. Given that clinical condition and geographical region will both impact the included population, we have not collated this information. However, we note that reporting of demographic characteristics varied. Whilst almost all studies reported core demographics (e.g., age, gender), reporting of wider factors that are of importance to personalised care was more limited; 68% report educational level, 37% participant ethnicity, 37% household income/employment status and 18% social deprivation (or related data, such as housing and insurance status).

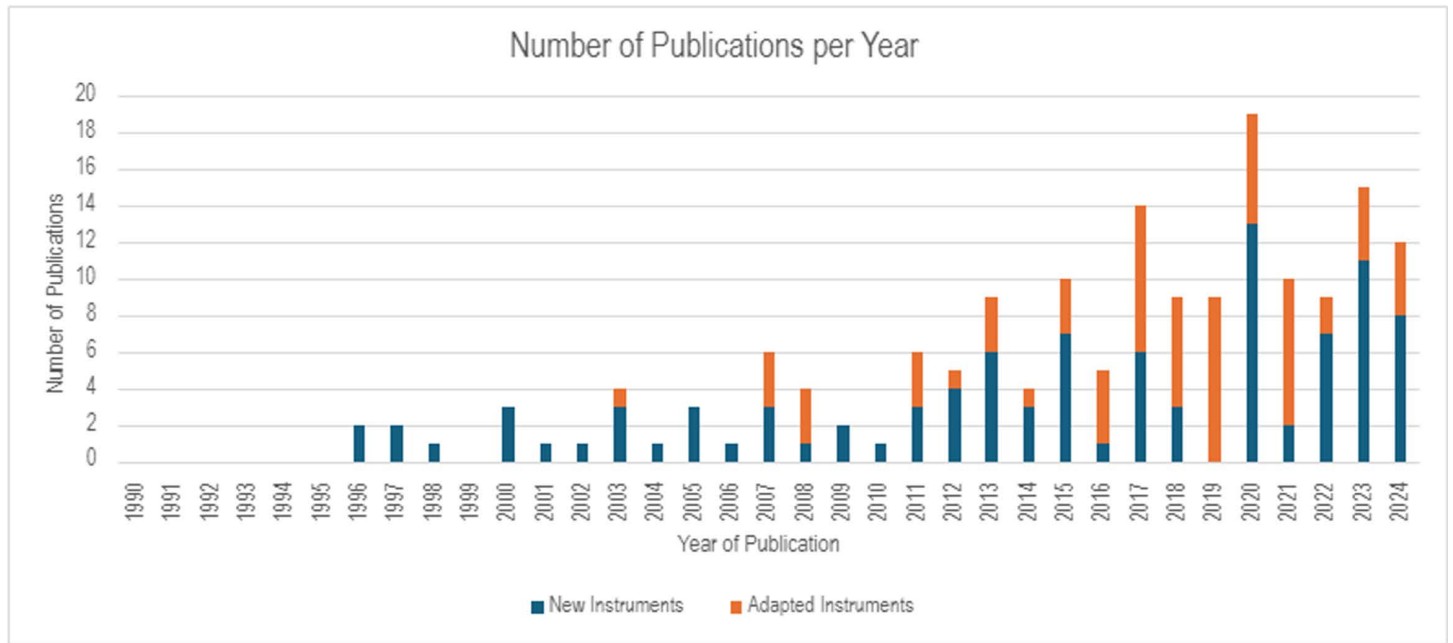

**Fig 2.  Publications over time.**

**Context.** Instruments were developed and evaluated in a wide range of healthcare settings and specialities (Table 1); most commonly in primary care. Many tools (81%) were designed for use in a specific clinical population; and whilst there were examples from a wide range of clinical specialities, instruments designed for use in diabetes care were particularly common (23%).

Instruments vary in both length and breadth; ranging from those purposively designed with clinical utility in mind, to those designed to give more comprehensive insights. The mean number of factors (domains) within an instrument was 4.3 (range 1–16; SD 2.7), and the mean number of items was 22.4 (range 2–91; SD 15.6).

**Context.** Of all identified instruments, most were unidimensional – focussing on one aspect of personalised care. The most common aspects were supported self-management (80%) and shared decision making (17%). We did not identify any instruments aimed specifically at evaluating the personalised care and support planning or social prescribing aspects of the personalised care model. Only a small number of instruments (n = 3) focussed on personalised care experiences more broadly. For example, The Kim Alliance Scale [22] is designed to evaluate the therapeutic alliance between patient and provider, and the Perceived Involvement in Care Scale [23] is designed to evaluate patient attitudes about illness and medical care. Both are grounded in an evaluation of patient-health care provider relationships and were developed to facilitate improvements in personalised interventions. We did not identify any multidimensional instruments that capture a range of personalised care domains.

**Content analysis of instruments.** Content analysis of individual instruments revealed a range of constructs. We categorised instruments broadly in terms of overall purpose: to be used pre-intervention to determine needs and to direct

**Table 1. Number of instruments aimed at service users, per clinical speciality.**

| Condition Group | Number of Instruments | % |
|---|---|---|
| Endocrine (diabetes) | 22 | 22.9 |
| Mixed Long-Term Conditions | 17 | 17.7 |
| Cancer | 7 | 7.3 |
| Cardiac | 8 | 8.3 |
| Musculoskeletal Conditions | 6 | 6.3 |
| Respiratory | 6 | 6.3 |
| Neurology | 5 | 5.2 |
| Mental Health | 5 | 5.2 |
| Renal | 5 | 5.2 |
| Geriatrics | 3 | 3.1 |
| Rehabilitation | 1 | 1.0 |
| Audiology | 1 | 1.0 |
| Pain | 1 | 1.0 |
| Hospital Discharge | 1 | 1.0 |
| Surgery | 1 | 1.0 |
| Haematology | 1 | 1.0 |
| Infectious Diseases | 1 | 1.0 |
| Vascular | 1 | 1.0 |
| Major Trauma | 1 | 1.0 |
| Gastroenterology | 1 | 1.0 |
| Rheumatology | 1 | 1.0 |
| General Health | 1 | 1.0 |
| Total | 96 | 100 |

treatment (42%); to be used post-intervention to evaluate outcome or experience (22%); or to be used flexibly across the time course of input, to tailor input, as well as to evaluate change (36%).

Whilst some instruments focus specifically on a single construct (e.g., knowledge), many evaluate a range. We found most tools included multiple choice or rating scales as response items, whilst some had open text answers that require a degree of analysis/coding, e.g., [24]

Content analysis generated six domains (Table 2). Some instruments focus on a single domain, and others cross several, but no instruments covered all six domains. These domains relate specifically to instruments that evaluate

**Table 2. Domains and Examples.**

| DOMAIN | EXAMPLE(s) |
|---|---|
| **Domain 1: Understanding the Person** | |
| 1.1 Values and Beliefs | *I have a firm belief which guides me for better diabetes control* [Character Strengths in Diabetes Self-Management Scale [25]] |
| 1.2 Priorities and Preferences | *I am confident with controlling my symptoms* [Chronic Hepatitis B Self-Management Scale [26]] |
| 1.3 Self-Efficacy and Confidence | *I have been asked about my values and traditions* [Patient Assessment of Care in Chronic Conditions [27]] |
| 1.4 Acceptance and Resilience | |
| **Domain 2: Understanding Capability** | |
| 2.1 Knowledge – Condition Specific | Knowledge: *During an asthma attack, the muscles around the air tubes tighten and the tubes become narrow (True/False)* [Asthma General Knowledge Questionnaire [28]] |
| 2.2 Knowledge – General | Skill: *(I am able to ensure) The solution bags are correctly attached and the tubes are correctly organized* [Self-Management Scale for Peritoneal Dialysis [29]] |
| 2.3 Skills – Physical | |
| 2.3 Skills – Psychological | |
| **Domain 3: Personalised Care Interventions** | |
| Interventions offered or accessed | *In the last 3 months, how often have you used community resources to help manage your illness such as senior centres, community centres, or mall walking programmes* [The Chronic Illness Resources Survey [30]]. |
| | *Attending support groups is an important part of my HIV management strategy* [HIV Self-Management Scale [30]] |
| **Domain 3: Understanding Behaviour** | |
| 4.1 Intent or Preparedness | Intent: *I realize now that it is time for me to come up with a better plan to cope with or manage my injury related problems* [Readiness to Engage in Self-Management after Acute Traumatic Injury [31]] |
| 4.2 Behavioural Action | Action – Condition Specific: *I check my blood sugar levels with care and attention* [Diabetes Self-Management Questionnaire [32]] |
| 4.2 Behavioural Impact | Action/Impact – General: *I do things to maintain a healthy weight* [Adult Epilepsy Self-Management Measurement Instrument [33]] |
| **Domain 4: Personalised Care Interventions** | |
| Interventions offered or accessed | *In the last 3 months, how often have you used community resources to help manage your illness such as senior centres, community centres, or mall walking programmes* [The Chronic Illness Resources Survey [34]]. |
| | *Attending support groups is an important part of my HIV management strategy* [HIV Self-Management Scale [30]] |
| **Domain 5: Experience** | |
| 5.1 Experience and relationship with healthcare team | *My HIV doctor and I have a good relationship* [HIV Self-Management Scale [30]] |
| | *I am comfortable suggesting treatment plan changes to my health care provider* [Diabetes Self-Management Instrument [35]] |
| 5.2 Experience and satisfaction of care overall | *I have problems with different healthcare providers not communicating with each other about my medical care* [Patient Experience with Treatment and Self-Management [36]] |
| **Domain 6: Wider Determinants of Health** | |
| 6.1 Social | *To what extent did cost or insurance make it hard to take best care of your diabetes?* [Barriers and Supports Evaluation [37]] |
| 6.2 Economic | |
| 6.3 Environmental | |

personalised care from a service user perspective. There were insufficient instruments to generate similar domains in relation to healthcare professionals.

**Domain 1: Understanding the Person** includes items that seek to understand the person (patient) and their unique sense of self; typically for the purpose of tailoring care interventions. Included within this domain are items that explore a person's values, beliefs, priorities and preferences; characterising individual aspects that are important to personalisation of care. Whilst these items may be influenced by external factors, they are inherently internal to a person. This domain also includes items relating to character strengths, exploring self-efficacy, confidence, acceptance and resilience. These items are potentially changeable and therefore understanding them is important not just for tailoring interventions, but also as the direct focus of those interventions, e.g., a self-management support programme aimed at building self-efficacy.

**Domain 2: Understanding Capability** includes items that explore a person's knowledge and skills, in relation to one or more component of personalised care. Knowledge was further categorised as general or condition specific; and skills were further categorised as physical or psychological. Included within psychological skills were executive functions, such as problem solving and proactivity.

**Domain 3: Understanding Behaviour** primarily includes items that ask about actions – i.e., what someone does and/or how often they do it. Behaviours were both general and condition specific. Whilst we acknowledge that many instruments exploring behaviour are grounded in behavioural change theory, for simplicity, we categorised behaviour related items into one of three groups. Behavioural intent, including preparedness or readiness for change (what someone plans to do); behavioural action, exploring what someone actually does (or perceives they do); and behavioural impact, including items that explore someone's ability to identify change, and/or insight into the outcome of their behaviours. As behaviour is influenced by both the person and their capability, this forms the central thread of our proposed framework.

**Domain 4: Personalised Care Interventions** includes items that explore what happens for someone; what interventions were given or what was offered by the service.

***Domain 5: Experience of Care*** explores a person's experience (or satisfaction) with healthcare; whether relating to a specific intervention, or their overall healthcare experience. This domain includes items about a person's relationship and trust with their healthcare practitioner, healthcare team, or the health system more broadly.

**Domain 6: Wider Determinants** covers any social, economic or environmental factors that may influence a person's mental or physical health.

The proposed schematic (Fig 3) illustrates the likely relationship between these domains. Factors relating to "*the person" and "capability" (Domain 1 and 2)* will influence an individual's *"behaviour" (Domain 3);* and these insights can be used to inform "*personalised care interventions*" (Domain 4), through needs based and tailored treatment plans. Equally, those interventions, if targeted correctly, should mediate behaviours, and may alter aspects of Domains 1 and 2 (e.g., increasing knowledge or improving self-efficacy). All factors impact "*experience*" (Domain 5). *Wider determinants of health* (Domain 6) sit around the outside, as wider contextual factors that can influence all aspects of the model.

## Discussion

To our knowledge, this is the first review to summarise the breadth and range of instruments available within the personalised care field.

Despite featuring heavily in healthcare policy across many parts of the world, the widespread implementation of personalised care is lagging. The ambition to embed personalised care within modern health systems is hindered by many challenges, including that of evaluation. This raises questions for how we effectively understand and evaluate a complex, nebulous construct, which is shaped by the ethos and culture of those delivering it and experienced uniquely by those receiving it.

It can be difficult to rate personalised care if patients (or indeed clinicians) lack a frame of reference, and therefore instruments need to be specific enough to elicit useful responses, but broad enough to capture insights into a concept

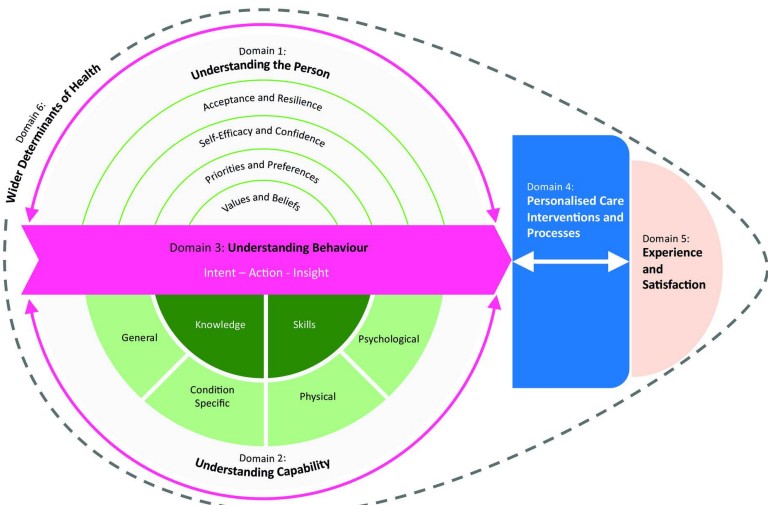

**Fig 3. Personalised Care Evaluation Domains (PCED-6).**

that is by nature, underpinned by an individual's values and preferences [38]. This review identified a large volume of instruments, designed to support personalised care evaluation in some way. Many of the instruments identified were designed to understand a single construct of care (e.g., supported self-management), to do this at an individual level (e.g., patients) and in a specific clinical population (e.g., diabetes). For clinicians looking to utilise a standardised instrument for a specific purpose, there are many to choose from. Such tools can aid with tailoring treatment for an individual, and if used well, can act as a clinical prompt and conversation starter to facilitate a more personalised approach to care. Such assessment scales can help clinicians to understand the status and ability of patients, to formulate a common understanding, and consequently provide more individualised and effective support to meet individual need [39]. At the latter end of the patient experience, outcomes from specific personalised care initiatives can be evaluated using patient reported outcome and experience measures (PROMs and PREMs); allowing patients to report how they function or feel with respect to their health and wellbeing, or their healthcare experience [Person-Centered Outcome Measures - NCQA].

The PCED-6 framework developed through this review, highlights the range of constructs that could form part of a personalised care evaluation. Which ones are relevant, is dependent on the context and purpose of any given evaluation. Future development of the framework could consider the weighting of each domain, ideally in a dynamic manner that is tailored to the importance of each domain to the individual person and situation. When selecting an instrument to use in clinical practice or research, mapping to this framework will aid contextualisation and provide clarity on what is and isn't being considered within the chosen instrument.

Most of the instruments identified through this scoping review, focus on either supported self-management or shared decision making. As self-management encompasses all the actions a person takes daily to manage symptoms, avoid relapse and optimise well-being (Lorig and Holman, 2003) [40] , it is not surprising the instruments measuring self-management had a strong focus on behaviours, as well as the knowledge, skills and confidence to sustain these behaviours. Instruments that include a range of factors, rather than evaluating just one construct (e.g., knowledge), are likely to give insights that are more impactful for clinical practice. For SDM, previous research has classified instruments into three categories: tools that capture decision antecedents (e.g., role preference), scales that describe the decision-making process, and instruments that assess decision outcomes (i.e., decision quality, decisional conflict, regret, knowledge) [10]. We found tools that focus on SDM to broadly fit with these categories. Overall, content of the identified SDM

tools was homogenous, likely because SDM is a well-defined construct with an underpinning theoretical basis. One limitation of the instruments for SDM, is that most were designed to evaluate a single interaction or decision; with an over-emphasis on medical decision making. In clinical practice, decision making is not (and should not) always be neatly bound into one interaction. A heavy emphasis on measuring single events in healthcare has been noted elsewhere [41], with most PREMS related to short-term care episodes, largely in the hospital setting, and limited to a singular disease focus. Future PREM development should aim to capture experiences of the continuity and coordination within and between health care services and providers [41], a notion that is supported throughout this current review. Whilst there is value in having instruments that focus on specific interventions for personalised care delivery (i.e., SDM and SSM), our review highlights a lack of tools that account for the complex and interacting nature of personalised care, and how this is delivered across systems and over time.

The context in which personalised care is implemented is changing, which in time, may impact clinical and research utility of instruments. For example, some instruments include specific questions relating to digital confidence or competence [42]. Instruments with very specific questions about technical knowledge or skills, in relation to a specific condition, are likely to become outdated as clinical care evolves.

Many authors cite that no tools exist to assess self-management capability as the reason for tool development, yet this review found a plethora of instruments exist. There is a risk of oversaturation of condition specific instruments in this field, and efforts should shift focus to how instruments can be integrated within clinical pathways, to embed and improve personalised care delivery. Furthermore, it is known that the compound effects of multiple long-term conditions are an independent barrier to self-management [43,44]; instruments designed to understand the common barriers, and the cumulative impacts for those living with multiple long-term conditions, are essential.

This review did not identify any instruments designed to understand personalised care more broadly. As such, there is no universally accepted way to understand the translation of policy to practice at a system or national level, and in a way that captures the increasing complexity of health systems, or the challenge of multimorbidity. An excess of definitions contributes to this challenge and has led to the development and adaption of many tools. Capturing all aspects of personalised care within a single tool presents conceptual challenges, reflecting the potential for multi-instrument approaches. The PCED-6 framework could be used to guide such approaches, aiding instrument selection in line with the intended purpose of the evaluation. Our review also highlights a lack of instruments that consider multiple perspectives. We identified very few instruments that were targeted at understanding the perspectives of health care professionals, or informal caregivers/family members. No instruments were specifically designed for system leaders, or those in senior leadership or policy roles. Evidence suggests that what people/patients want, and what clinicians think they want, can be very different [27]. At a delivery level, achieving personalised care requires a horizontal balance of power, and is directly influenced by the attitudes and approaches of healthcare practitioners [45]. Wider contextual factors, at an organisational and system level, also act and interact as barriers or facilitators to personalised care delivery [46]. Yet current evaluation instruments focus almost exclusively on the patient perspective. This misses a vital opportunity to identify, describe and compare wider factors that govern implementation, and is at odds with the fundamental notion that successful personalised care delivery requires a whole system approach. This is an important area for future international research.

## Strengths, limitations and challenges

Defining personalised care as a concept and setting the boundaries for inclusion within this scoping review, was a challenge. We used the NHS England Comprehensive Model of Personalised Care to develop our search terms, keeping our lens purposively broad. To ensure there was a degree of focus to the review, and the size remained manageable, we excluded instruments that evaluate constructs related to personalised care, but that are not personalised care per se – for example, scales for measuring patient activation and self-efficacy. Arguably, these concepts could be the most useful ones to measure in clinical practice, and their exclusion does not reflect their importance.

It was not within the scope of this review to report the reliability or validity of the included measures. Given the overwhelming number of instruments identified, future work to clarify and report the psychometric properties and clinical application of various instruments, could help clinicians in their selection and use. Importantly, the instrument is only the start, and how it is used constructively to inform actions that improve personalisation of care, is crucial.

Whilst we have mapped the development of instruments around the globe, we have only included studies published in the English language and accept this may skew the true geographical picture.

## Conclusions

This review provides a comprehensive overview of the current body of evidence relating to the evaluation of personalised care delivery. Personalised care is a complex intervention, made up of multiple interacting behaviours. Its principles are applicable in a broad range of settings, for a broad range of purposes – benefitting not just individuals, but also the wider health system. The implementation of personalised care is widely accepted as being reliant on a shift in attitudes and culture within healthcare and is therefore influenced by a range of stakeholders. This complexity makes the evaluation and measurement of personalised care globally challenging. Nonetheless, evaluation is an important pre-requisite to improving delivery and meeting the ambitious challenge of operationalising personalised care across health systems and countries. Through an improved understanding of current methods to objectively and consistently evaluate the delivery of care that is personalised, areas for improvement can be identified and targeted, and change can be monitored.

We found a wide range of instruments that have been developed and tested within a range of populations and settings. However, this review confirms that we currently lack instruments that a) can be broadly applied, including to understand the experiences of people living with multiple long-term conditions and b) provide insights into the multiple perspectives (e.g., healthcare professionals, family carers) that have a role in personalised care delivery. Future studies should focus efforts towards the development and use of single or multi-instruments that address these gaps, allowing for comparison and shared learning across and between services and health systems. Future work should also focus on *how* instruments are used to improve personalised care delivery, particularly through a less siloed, multimorbidity lens.

Contributions to the literature:

- There are a wide range of published instruments designed to inform, understand or evaluate the delivery of personalised care interventions.

- Most instruments focus on one element of personalised care (e.g., shared decision making) and are designed for a specific clinical population (e.g., diabetes).

- Across all available instruments, a broad range of constructs are evaluated. These have been summarised as six domains and used to propose the Personalised Care Evaluation Domains model – PCED-6.

- Future work is required to understand if and how standardised instruments should be operationalised, to improve personalised care delivery in practice.

## Supporting information

**S1 File. Search Strategy.**
(DOCX)

**S2 File. Summary of Included Papers.**
(DOCX)

 

## Author contributions

**Conceptualization:** Louise Johnson, Beth Clark, Lyndsay Court, Hayden Kirk, Matthew Wood.

**Data curation:** Louise Johnson, Beth Clark, Lyndsay Court, Hayden Kirk, Matthew Wood, Sharon Jackson.

**Formal analysis:** Louise Johnson, Mari Carmen Portillo.

**Methodology:** Louise Johnson, Mari Carmen Portillo.

**Project administration:** Louise Johnson.

**Supervision:** Mari Carmen Portillo.

**Visualization:** Louise Johnson, Beth Clark, Hayden Kirk, Matthew Wood.

**Writing – original draft:** Louise Johnson.

**Writing – review & editing:** Beth Clark, Lyndsay Court, Hayden Kirk, Matthew Wood, Sharon Jackson, Mari Carmen Portillo.

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
