## [Decision Letter · Decision Letter 0]

PONE-D-24-60456What instruments are available to aid or evaluate personalised care delivery, from the perspectives of healthcare practitioners and service users? A narrative scoping review.PLOS ONE

Dear Dr. Johnson,

Thank you for submitting your manuscript to PLOS ONE. After careful consideration, we feel that it has merit but does not fully meet PLOS ONE’s publication criteria as it currently stands. Therefore, we invite you to submit a revised version of the manuscript that addresses the points raised during the review process.

**ACADEMIC EDITOR: **

Thank you for submitting your manuscript to *PLOS ONE* . After careful consideration and review, I believe your manuscript addresses an important and timely topic. While it has merit, it does not yet fully meet *PLOS ONE* ’s publication criteria in its current form. Therefore, I am inviting you to submit a **major revision** that addresses the points raised during the peer review process.

When submitting your revision through Editorial Manager , please include the following:

A **rebuttal letter** that responds point-by-point to each comment from the academic editor and reviewers. Upload this as *Response to Reviewers* .A **marked-up version** of your revised manuscript showing all changes made (e.g., with Track Changes). Upload this as *Revised Manuscript with Track Changes* .A **clean version** of your revised manuscript without tracked changes. Upload this as *Manuscript* .If applicable, an updated **financial disclosure statement** in your cover letter.

 You may also consider depositing any protocols referenced in your review process to protocols.io , where they can be assigned a DOI and cited independently. For more details about submitting Lab Protocol articles, visit: https://plos.org/protocols .

We look forward to receiving your revised manuscript.

Kind regards,

Sefki Kolozali

Academic Editor

PLOS ONE

Journal Requirements:

2.  We are unable to open your figure file [Fig 2 Publications per year.eps]. Please kindly revise as necessary and re-upload.

Reviewers' comments:

Reviewer's Responses to Questions

**Comments to the Author**

1. Is the manuscript technically sound, and do the data support the conclusions?

Reviewer #1: Yes

Reviewer #2: Yes

2. Has the statistical analysis been performed appropriately and rigorously? 

Reviewer #1: Yes

Reviewer #2: I Don't Know

3. Have the authors made all data underlying the findings in their manuscript fully available?

Reviewer #1: Yes

Reviewer #2: Yes

4. Is the manuscript presented in an intelligible fashion and written in standard English?

Reviewer #1: Yes

Reviewer #2: Yes

5. Review Comments to the Author

Reviewer #1: The authors provided a comprehensive overview of current personalised care delivery and described the existing tools, instruments or methods for assessing, evaluating or measuring personalised care from the perspectives of healthcare practitioners and/or service users. This is an important review for the field of personalised care, as a well-established personalised care will benefit not only individuals, but also the wider health system.

The manuscript is well structured and well-written. It also proposed a framework of six domains for content analysis of health care instruments. The framework is a sensible attempt to evaluate the personalised care instruments. My suggestions are:

1 the Figure 2 is at low resolution and hard to read. It will be easier to understand the figure if move text "domain 3: understanding behaviour" into the magenta bar.

2 A weighing system of the six domains maybe introduced when evaluating the instruments. The importance of the six domains may vary depends on the needs of individuals.

Reviewer #2: The work provides a valuable contribution to the field, offering a comprehensive overview of existing instruments and proposing a conceptual framework. However, there are several areas that require attention to strengthen the manuscript.

While the review is thorough, the narrative occasionally loses focus. The introduction and discussion sections could benefit from more concise and targeted language to better guide the reader through your objectives and findings. For instance, the introduction could more sharply delineate the gap in the literature that your review addresses.

The methodology section is detailed, but the exclusion criteria, particularly the decision to exclude instruments measuring constructs like self-efficacy and patient activation, need more robust justification given their relevance to personalised care.

While the iterative refinement of the data extraction process is commendable, it would be useful to clarify how inter-rater reliability was ensured during study selection and coding. Did reviewers use any specific metrics (e.g., Cohen’s kappa) to measure agreement?

The discussion section effectively summarises the findings but could delve deeper into the implications of the identified gaps. For example, the lack of instruments that consider multiple perspectives (e.g., healthcare professionals, caregivers) is a significant limitation that warrants more extensive discussion regarding its impact on the field.

The manuscript notes that no single instrument captures all aspects of personalised care. Could this indicate a need for a multi-instrument approach rather than a single new tool?

The conclusion hints at future research needs but could be more specific. Highlighting particular areas where new instrument development is most urgently needed, especially in the context of multimorbidity, would provide a clearer roadmap for future studies.

There are instances where the language could be more polished, such as: "prevenance" (line 67) should likely be "prevalence.", "Personlised" (figure 3 legend) should be "Personalised.". Phrases like "a plethora of instruments" could be replaced with more precise language. Additionally, varying sentence length and structure would improve readability and engagement, particularly in the discussion section.

Overall, your manuscript is a strong foundation that, with some refinement, can make a significant impact on the field of personalised care evaluation. I look forward to seeing the revised version.

6. PLOS authors have the option to publish the peer review history of their article (what does this mean? ). If published, this will include your full peer review and any attached files.

**Do you want your identity to be public for this peer review?** For information about this choice, including consent withdrawal, please see our Privacy Policy .

Reviewer #1: No

Reviewer #2: No

---

## [Author Response · Author response to Decision Letter 1]

14 Apr 2025

Thank you for inviting us to revise and re-submit this paper, describing the range of instruments available to inform, evaluate or measure personalised care delivery. We appreciate the valuable feedback offered by the reviewers, which has allowed us to further strengthen the paper.

We have responded to each comment raised by the editor and the reviewers in the table below. Each point has been addressed, and the amendments are highlighted as tracked changes in the uploaded manuscript.

Reviewer 1: The Figure 2 is at low resolution and hard to read. It will be easier to understand the figure if move text "domain 3: understanding behaviour" into the magenta bar

Response: Many thanks for highlighting this. We have made the suggested text change to the figure and have uploaded a new version in higher resolution.

Reviewer 1: A weighing system of the six domains maybe introduced when evaluating the instruments. The importance of the six domains may vary depends on the needs of individuals

Response: Thank you for this suggestion. We have considered the weighting of domains, but do not feel it would be within the scope of this review, to make recommendations on this. We have added a line into the second paragraph of the discussion, to highlight this as a concept for future consideration.

“Future development of the framework could consider the weighting of each domain, ideally in a dynamic manner that is tailored to the importance of each domain to the individual person and situation”

Reviewer 2: While the review is thorough, the narrative occasionally loses focus. The introduction and discussion sections could benefit from more concise and targeted language to better guide the reader through your objectives and findings. For instance, the introduction could more sharply delineate the gap in the literature that your review addresses

Response: We have made some changes to both the introduction and the discussion to aid conciseness and focus. We have also added the following text into the second paragraph to add clarity about the gap we are addressing:

“While there are various instruments available in the field of personalised care, there is a lack of guidance on how to effectively select and utilise these tools. Enhancing this understanding is essential to improving clinical delivery and systematic implementation. This scoping review will compile and categorise the existing instruments, laying the groundwork for a deeper understanding of their role in personalised care delivery, and identifying areas that require further research.”

Reviewer 2: The methodology section is detailed, but the exclusion criteria, particularly the decision to exclude instruments measuring constructs like self-efficacy and patient activation, need more robust justification given their relevance to personalised care

Response: Thank you for this comment. Defining the scope of this review was challenging, given the wide range of constructs that are linked to personalised care. We decided to exclude these types of instruments as they are related to personalised care but also existed independently of it. There is a significant volume of instruments in these wider fields (e.g. self-efficacy scales). Including them would risk the scale of the review becoming unmanageable; but also loosing focus. These concepts were identified within our content analysis of the included tools, where they were sitting within an instrument that evaluated a wider range of constructs. Therefore, their role in personalised care is acknowledged through the results of this paper, but instruments that were solely designed to evaluate one characteristic, were excluded. We have added clarity to the text regarding this:

“Instruments that sought to specifically and solely understand personal characteristics, such as self-efficacy, coping behaviours or activation, were excluded from the review. Whilst we recognise these are important characteristics that influence personalised care, they also play an independent role in overall health outcomes, beyond personalised approaches. To maintain a direct focus on personalised care delivery, we only included instruments that explored two or more personal characteristics, with the explicit intention of informing personalised care delivery. Given the significant number of related constructs and the high volume of instruments in these related fields, this was also important to maintain a realistic scope for the review.”

Reviewer 2: While the iterative refinement of the data extraction process is commendable, it would be useful to clarify how inter-rater reliability was ensured during study selection and coding. Did reviewers use any specific metrics (e.g., Cohen’s kappa) to measure agreement?

Response: We did not use any statistical methods to establish inter-rater reliability; which we believe is acceptable for a scoping review of this nature.

Reviewer 2: The discussion section effectively summarises the findings but could delve deeper into the implications of the identified gaps. For example, the lack of instruments that consider multiple perspectives (e.g., healthcare professionals, caregivers) is a significant limitation that warrants more extensive discussion regarding its impact on the field.

Response: Thank you for highlighting this. We have reviewed the discussion and added depth to aspects, including the point raised about multiple perspectives.

The manuscript notes that no single instrument captures all aspects of personalised care. Could this indicate a need for a multi-instrument approach rather than a single new tool?

Response: We agree, and have added a sentence within the discussion to reflect this:

“Capturing all aspects of personalised care within a single tool presents conceptual challenges, reflecting the potential for multi-instrument approaches. The PCED-6 framework could be used to guide such approaches, aiding instrument selection in line with the intended purpose of the evaluation”.

Reviewer 2: The conclusion hints at future research needs but could be more specific. Highlighting particular areas where new instrument development is most urgently needed, especially in the context of multimorbidity, would provide a clearer roadmap for future studies

Response: The following changes have been made to the conclusions to address this feedback:

“We found a wide range of instruments that have been developed and tested within a range of populations and settings. However, this review confirms that we currently lack instruments that a) can be broadly applied, including to understand the experiences of people living with multiple, long-term conditions and b) provide insights into the multiple perspectives (e.g. healthcare professionals, family carers) that have a role in personalised care delivery. Future studies should focus efforts on the development and use of single or multi-instruments that address these gaps, allowing for comparison and shared learning across and between services and health systems. This is increasingly important given the growing number of people living with multimorbidity, and recognition of the burden of this for healthcare delivery worldwide. Future work should also focus on how instruments are used to improve personalised care delivery, particularly through a less siloed, multimorbidity lens.”

Reviewer 2: There are instances where the language could be more polished, such as: "prevenance" (line 67) should likely be "prevalence.", "Personlised" (figure 3 legend) should be "Personalised.". Phrases like "a plethora of instruments" could be replaced with more precise language. Additionally, varying sentence length and structure would improve readability and engagement, particularly in the discussion section

Response: Thank you – we hope we have addressed this throughout.

---

## [Decision Letter · Decision Letter 1]

What instruments are available to aid or evaluate personalised care delivery, from the perspectives of healthcare practitioners and service users? A narrative scoping review.

PONE-D-24-60456R1

Dear Dr. Johnson

We’re pleased to inform you that your manuscript has been judged scientifically suitable for publication and will be formally accepted for publication once it meets all outstanding technical requirements.

Kind regards,

Sefki Kolozali

Academic Editor

PLOS ONE

Additional Editor Comments (optional):

Reviewers' comments:

Reviewer's Responses to Questions

**Comments to the Author**

1. If the authors have adequately addressed your comments raised in a previous round of review and you feel that this manuscript is now acceptable for publication, you may indicate that here to bypass the “Comments to the Author” section, enter your conflict of interest statement in the “Confidential to Editor” section, and submit your "Accept" recommendation.

Reviewer #1: All comments have been addressed

Reviewer #2: All comments have been addressed

2. Is the manuscript technically sound, and do the data support the conclusions?

Reviewer #1: Yes

Reviewer #2: Yes

3. Has the statistical analysis been performed appropriately and rigorously? 

Reviewer #1: N/A

Reviewer #2: N/A

4. Have the authors made all data underlying the findings in their manuscript fully available?

Reviewer #1: Yes

Reviewer #2: Yes

5. Is the manuscript presented in an intelligible fashion and written in standard English?

Reviewer #1: Yes

Reviewer #2: Yes

6. Review Comments to the Author

Reviewer #1: The authors have addressed most of the comments raised in their revision. They also explained the reason why one of the suggestions were not included in the revision. Based on the quality of the revised manuscript. I think it is acceptable for publication.

Reviewer #2: (No Response)

7. PLOS authors have the option to publish the peer review history of their article (what does this mean? ). If published, this will include your full peer review and any attached files.

**Do you want your identity to be public for this peer review?** For information about this choice, including consent withdrawal, please see our Privacy Policy .

Reviewer #1: No

Reviewer #2: **Yes: ** Basma A. Al-Ghali

---

## [Editor Report · Acceptance letter]

PONE-D-24-60456R1

PLOS ONE

Dear Dr. Johnson,

I'm pleased to inform you that your manuscript has been deemed suitable for publication in PLOS ONE. Congratulations! Your manuscript is now being handed over to our production team.

Kind regards,

on behalf of

Dr. Sefki Kolozali

Academic Editor

PLOS ONE